# REDUCING COMPLEXITY OF FORCE-DIRECTED GRAPH EMBEDDING

## ABSTRACT

Graph embedding is a critical pre-processing step that maps elements of a graph network, such as its nodes or edges, to coordinates in a $d$-dimensional space. The primary goal of the embedding process is to capture and preserve various features of the graph network, including its topology and node attributes, in the generated embedding. Maintaining these graph features in the embedding can significantly enhance the performance of the downstream machine learning tasks. In this work, we introduce a novel family of graph embedding methods that leverage kinematics principles within a spring model and $n$-body simulation framework to generate the graph embedding. The proposed method differs substantially from state-of-the-art (SOTA) methods, as it does not attempt to fit a model (such as neural networks) and eliminates the need for functions such as message passing or back-propagation. Instead, it aims to position the nodes in the embedding space such that the total net force of the system is reduced to a minimal threshold, resulting in the system reaching an equilibrium state. The spring model is designed as a linear summation of non-linear force functions, with the shortest-path distance serving as the adjusting parameter for the force factor between each node pair, and therefore, inducing the graph topology in the force functions. In this work, we attempted to reduce the complexity of the original algorithm from $\log(n^2)$ to $n \log(n)$, while maintaining the performance metrics at a competitive level. The proposed method is intuitive, parallelizable, and highly scalable. While the primary focus of this work is on the feasibility of the force-directed approach, the results in unsupervised graph embeddings are comparable to or better than SOTA methods, demonstrating its potential for practical applications.

## 1 INTRODUCTION

Graphs have become the go-to data structure for representing complex systems and relationships between data entities Wu et al. (2020); Li et al. (2021). A graph, denoted as $G(\mathcal{V}, \mathcal{E})$ is comprised of a set of $n$ nodes denoted as $\mathcal{V} = \{u_1, u_2, ..., u_n\}$, and the set of edges connecting some node pairs and denoted as $\mathcal{E} = \{(u_i, u_j)\}$, such that $u_i, u_j \in \mathcal{V}$. Graph embedding is the task of mapping graph elements down to a vector space with $d$ dimensions, such that $d \ll n$. It has gained significant attention in recent years due to the emergence of big data and advancements in machine learning and deep learning techniques for graph representation learning.

In this paper, we propose a new family of graph embedding methods, dubbed Force-Directed embedding, based on the principles of motion physics and Newton's second law and a $n$-body simulation scheme. By treating graph nodes as objects with mass that exert forces on each other and using shortest-path distance between each pair as a parameter for determining the magnitude of the force factor, we aim to map the graph elements to a vector space while preserving the graph's topological features. The force-directed spring model employed in this approach converges to a state where the vector representation of nodes in the embedding space reflects their relative distances in the graphs as well as various graph features such as nodes clusters.

Unlike the conventional methods, we don't fit a function based on a loss metric. Instead, we deploy an iterative process to calculate the gradient of embedding, and update the node embeddings. Therefore, the proposed method does not need backward pass to fit parameters of a function and provides a performance advantage.

The proposed paradigm has an intuitive nature and is highly parallelizeable. By leveraging well-established principles from physics and the mathematics of the $n$-body problem, this approach explores a new avenue for graph embedding. A proof of convergence for this Force-Directed method was proposed in Lotfalizadeh & Al Hasan (2024) which also indicated the constraints for convergence. In this paper, we extend the work at Lotfalizadeh & Al Hasan (2023) and reduce the complexity of the algorithm by limiting the number of calculations to the forces between node pairs in a limited subset, while maintaining the quality of the embedding at a competitive level.

The remainder of this paper is organized as follows: Section II discusses related work, Section III presents the proposed Force-Directed graph embedding paradigm, Section IV details the experimental setup and results, and Section V concludes the paper and outlines future research directions.

## 2 RELATED WORKS

Existing graph embedding techniques can be broadly categorized into several types based on their approach to capturing the structure and features of the graph. Walk-based methods, such as Deep-Walk Perozzi et al. (2014) and node2vec Grover & Leskovec (2016), generate embeddings by conducting random walks across the graph. Deep learning-based methods leverage graph neural networks (GNNs) to learn representations of graph vertices or entire graphs. Notable GNN approaches include Graph Convolutional Networks (GCNs) Kipf & Welling (2016a); Chen et al. (2020), Graph-SAGE Hamilton et al. (2017), Graph Attention Networks (GATs) Velickovic et al. (2017), and Variational Graph Auto-Encoders (VGAEs) Kipf & Welling (2016b). These methods incorporate both local graph topology and node features to learn expressive embeddings. Spectral-based methods Zhang et al. (2021); Li et al. (2018) aim to capture global graph properties into the node embeddings by utilizing the eigenvalues and eigenvectors of the graph Laplacian to embed nodes in a way that preserves global graph properties. Matrix factorization methods Qiu et al. (2018); Yang et al. (2008) capture the graph structure through decomposing the adjacency matrix or other matrix representations of a graph into lower-dimensional matrices. These methods aim to preserve node connectivity, community structure, and node centrality in the lower-dimensional representation.

Force-Directed approaches have been widely employed for graph visualization purposes Eades (1984); Fruchterman & Reingold (1991); Kamada et al. (1989). These algorithms model the graph as a physical system, where nodes are treated as particles and edges as springs or forces between the particles, aiming to find a layout that minimizes the energy of the system. Advancements in Force-Directed graph drawing algorithms Barnes & Hut (1986); Walshaw (2001); Hu (2005) have enabled the visualization of larger and more complex graphs while preserving aesthetic properties such as symmetry, uniform edge lengths, and minimal edge crossings.

## 3 OVERVIEW OF FORCE-DIRECTED FRAMEWORK

Force-Directed graph embedding is inspired by the principles of motion physics. In this approach, nodes are taken as objects with mass that can relocate in the embedding space under the influence of attractive and repulsive forces. Using kinematics equations and Newton's second law, one can derive the equation (equation 1) to calculate the gradient of embedding at each step. The details of the derivation are discussed in Lotfalizadeh & Al Hasan (2023; 2024). In this equation, $\mathbf{z}_u$ is the vector representation or embedding of node $u$ and $\mathrm{d}\mathbf{z}_u$ is the gradient of embedding. This gradient is calculated by diving the net force on node $u$, $\mathbf{F}_u$, by its mass. In this setting, the degree of a node is taken as its mass. We need to define and calculate the net force.

$$\mathrm{d}\mathbf{z}_u = \frac{\mathbf{F}_u}{\deg u} \tag{1}$$

Each pair of nodes can exert mutual forces on each other. The objective is to set up the force functions such that the exerted forces lead the system to an equilibrium state where the relative positions of nodes in the embedding space reflect their topological distances. As a result, it should also capture the topological features of the graph in a global perspective.

In the following subsections, the Force-Directed framework is concisely outlined

### 3.1 THE ALGORITHM

The procedure of Force-Directed graph embedding approach is outlined in Algorithm 1. The algorithm iteratively calculates the gradient of embedding for each node from the net force on it. Subsequently, it updates node embeddings.

---

**Algorithm 1** Force-Directed Graph Embedding

---

1: Obtain $h_{uv}$ (shortest-path distance) for all $u, v \in \mathcal{V}$
2: Let $h_{uv} = |\mathcal{V}|$ if $u$ and $v$ are disconnected
3: Randomly initialize $\mathbf{z}_u$ for all $u \in \mathcal{V}$
4: **while** $\sum_{u \in \mathcal{V}} \|\mathbf{F}_u\| > \epsilon$ **do**
5:      **for all** $u \in \mathcal{V}$ **do**                                  ▷ Calculate gradients
6:          $\mathrm{d}\mathbf{z}_u = \frac{\mathbf{F}_u}{\deg u}$
7:      **end for**
8:      **for all** $u \in \mathcal{V}$ **do**                                  ▷ Update embeddings
9:          $\mathbf{z}_u \leftarrow \mathbf{z}_u + \mathrm{d}\mathbf{z}_u$
10:     **end for**
11: **end while**

---

### 3.2 THE FORCE FUNCTIONS

The equation equation 2 outlines the net force on node $u$ as the normalized sum of forces exerted on it by all other nodes. In this equation, $\mathbf{F}_u$ is the net force on node $u$, $\mathbf{F}_{uv}$ is the force exerted from node $v$ to $u$, and $\kappa$ is the normalization factor for controlling the convergence properties of the system.

$$\mathbf{F}_u = \sum_{v \in \mathcal{V}} \kappa \mathbf{F}_{uv} \tag{2}$$

The force between a pair of nodes $u$ and $v$ is defined as equation 3 where $\mathbf{z}_u$ and $\mathbf{z}_v$ are the vector representations of nodes $u$ and $v$ in the embedding space, and $f_{uv}$ is the force factor which is a scalar function of the Euclidean distance between the two nodes, with the shortest path distance, $h_{uv}$, as a constant. The force factor determines the magnitude and polarity of the force along the unit direction from $\mathbf{z}_u$ to $\mathbf{z}_v$, or $\frac{\mathbf{z}_v - \mathbf{z}_u}{\|\mathbf{z}_v - \mathbf{z}_u\|}$. A positive force factor makes node $u$ attract towards $v$, and a negative force factor makes $u$ move in the other direction. For the sake of brevity, we let $\mathbf{z}_{uv} = \mathbf{z}_v - \mathbf{z}_u$.

$$\mathbf{F}_{uv} = f_{uv}(\|\mathbf{z}_{uv}\|) \frac{\mathbf{z}_{uv}}{\|\mathbf{z}_{uv}\|} \tag{3}$$

To ensure the convergence of the Force-Directed system, the constraint in equation 4 needs to be satisfied. Letting $\kappa = \frac{1}{|\mathcal{V}|}$, a more simplified and constricted constraint can be derived as equation 5. This constraint guarantees the existence of an equilibrium point where the net forces on all nodes reach zero, as proven using Brouwer's fixed-point theorem Lotfalizadeh & Al Hasan (2024). Figure 1a depicts the upper bound $y = x$ for any force factor as a constraint.

A possible function that satisfies the constraints is depicted in 1a and provided in equation 6, with $x \in \mathbb{R}^{\geq 0}$ as the Euclidean distance and $h_{uv}$ as the shortest-path distance. The positive and negative components of this function work as the attractive and repulsive force factors. Increase of $x$, increases and decreases the attractive and repulsive components, respectively. On the other hand, $h_{uv}$ has the opposite effect.

$$\lim_{x \to \infty} \frac{\kappa \sum_{v \in \mathcal{V}} f_{uv}(x)}{x} < 1 \tag{4}$$

$$\lim_{x \to \infty} \frac{f_{uv}(x)}{x} < 1 \tag{5}$$

$$f_{uv}(x) = x \cdot e^{-h_{uv}} - h_{uv} \cdot e^{-x} \tag{6}$$

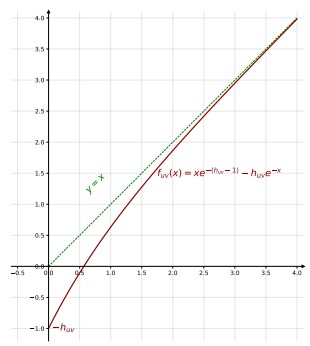 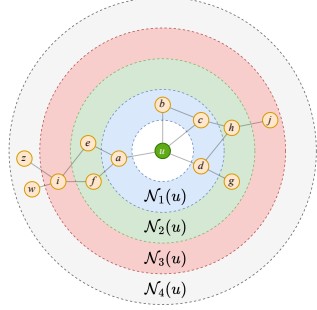

(a) The constraint equation 5 suggests that the force factor be a monotonically increasing function with upper bound $y = x$.

(b) The $k$-hop neighborhoods used for optimizing the force functions. Nodes in a same-color band pertain to the same neighborhood set.

Figure 1: The figures depicting the constraint and $k$-hop neighborhoods.

## 3.3 OPTIMIZING WITH $k$-HOP NEIGHBORHOOD SETS

The force factor indicated in equation 6 suffers from slow convergence rate and unsatisfying embedding quality. In Lotfalizadeh & Al Hasan (2023), the force factor function was optimized by first splitting the force factor into repulsive and attractive components and then, summing the average of attractive forces from each $k$-hop neighborhood of $u$. The $k$-hops neighborhood of $u$ is the set of nodes at exactly $k$ hops away from $u$, and is denoted and defined as $\mathcal{N}_k(u) = \{v \in \mathcal{V} \mid h_{uv} = k\}$. equation 7 depicts the net force as sum of attractive and repulsive forces. equation 8 shows the net attractive force on $u$ as the sum average of attraction from nodes in each $h$-hop neighborhood, with $h$ ranging from 1 to a maximum value $\max h_u = \max d(u, w), w \in \mathcal{V}$. equation 9 shows the net repulsive force as a simple summation of repulsion from all nodes. The parameters $k_1$, $k_2$, $k_3$, and $k_4$ adjust the effect of Euclidean and shortest path distances on the forces. Figure 1b shows the $k$-hop neighborhoods of $u$ in each colored band.

$$\mathbf{F}_u = \mathbf{F}_u^{(a)} + \mathbf{F}_u^{(r)} \tag{7}$$

$$\mathbf{F}_u^{(a)} = \sum_{h=1}^{\max h_u} \frac{1}{|\mathcal{N}_h(u)|} \sum_{v \in \mathcal{N}_h(u)} k_1 \cdot \|\mathbf{z}_v - \mathbf{z}_u\| \cdot e^{-k_2 \cdot (h_{uv}-1)} \cdot \frac{\mathbf{z}_v - \mathbf{z}_u}{\|\mathbf{z}_v - \mathbf{z}_u\|} \tag{8}$$

$$\mathbf{F}_u^{(r)} = \sum_{v \in \mathcal{V}} k_3 \cdot h_{uv} \cdot e^{-k_4 \cdot \|\mathbf{z}_v - \mathbf{z}_u\|} \cdot \frac{\mathbf{z}_v - \mathbf{z}_u}{\|\mathbf{z}_v - \mathbf{z}_u\|} \tag{9}$$

## 4 THE PROPOSED METHOD FOR REDUCING THE COMPLEXITY

In this section, we present a stochastic method to reduce the complexity of Force-Directed method by limiting the number of force computations to a limited subset of node pairs. While grouping the nodes into $k$-hop neighborhoods enhances performance metrics, the process is still computationally expensive at $O(n^2)$. The proposed method decreases the complexity of the Force-Directed embedding method to $O(n\Delta(G)^k + n \log n)$, such that $k \in \{1, 2, 3, 4\}$.

### 4.1 THE IDEA

The proposed idea is to calculate the net force on node $u$ from a limited number of nodes, denoted here by $\mathcal{V}(u)$. This set is comprised of the $k$-ball centered at $u$ and a maximum of $m$ nodes beyond the $k$-ball. In other words, we calculate the forces from all the nodes at a maximum of $k$-hops distance from $u$, and $m$ random nodes at a further distance. In our experiments, we let $m = O(\log n)$. In equation 10, $\mathcal{B}_k(u)$ is the $k$-ball centered at $u$, and $\mathcal{R}_{m,k}$ is a set of a maximum of $m$ nodes, sampled randomly from $\mathcal{V}$, without substitution, and not in the $k$-ball.

$$\mathcal{V}(u) = \mathcal{B}_k(u) \cup \mathcal{R}_{c,k}(u) \tag{10}$$

$$\mathcal{B}_k(u) = \{v \in \mathcal{V} \mid d(u,v) \leq k\} \tag{11}$$

$$\mathcal{R}_{m,k}(u) = \{v_1, \ldots, v_m\} \subseteq \{v \in \mathcal{V} \mid d(u,v) > k\} \tag{12}$$

We update the force functions by considering attraction on $u$ only from the nodes in $k$-ball set, as in equation 13, and considering the repulsion from nodes in $\mathcal{V}(u)$, as defined in equation 14.

$$\mathbf{F}_u^{(a)} = \sum_{h=1}^{k} \frac{1}{|\mathcal{N}_h(u)|} \sum_{v \in \mathcal{N}_h(u)} k_1 \cdot \|\mathbf{z}_{uv}\| \cdot e^{-k_2 \cdot (h_{uv}-1)} \cdot \frac{\mathbf{z}_{uv}}{\|\mathbf{z}_{uv}\|} \tag{13}$$

$$\mathbf{F}_u^{(r)} = \sum_{v \in \mathcal{V}(u)} k_3 \cdot h_{uv} \cdot e^{-k_4 \cdot \|\mathbf{z}_{uv}\|} \cdot \frac{\mathbf{z}_{uv}}{\|\mathbf{z}_{uv}\|} \tag{14}$$

### 4.2 THE RATIONALE

The logic behind definitions for $\mathcal{V}(u)$ and the updated force functions is that to reflect the topolofy of the graph in local and global granularity. By enforcing attraction and repulsion on $u$ from all the nodes in a close proximity, we can reflect the local topology of the graph in a short Euclidean proximity. On the other hand, enforcing repulsion from distant nodes helps with avoiding folding of the distant clusters into close vicinity of $u$ and reflecting global structure of the graph.

!!! A PICTURE TO BE INSERTED for CAMERA READY !!!

## 5 EXPERIMENTAL RESULTS

### 5.1 DATASETS AND BASELINE METHODS

To rigorously evaluate the efficacy of our proposed Force-Directed Graph Embedding method, we employ a diverse set of benchmark datasets widely recognized in the graph representation learning community. These datasets span various domains and exhibit different structural properties, enabling a comprehensive assessment of our method's performance across different graph types.

- **Cora** Sen et al. (2008): A citation network comprising 2,708 scientific publications categorized into seven classes, interconnected by 5,429 citation links.

- **CiteSeer** Sen et al. (2008): Another citation network consisting of 3,312 scientific publications across six topics, with 4,732 inter-publication citations.

- **PubMed Diabetes** Namata et al. (2012): A specialized dataset containing 19,717 diabetes-related scientific publications from the PubMed database, classified into three categories and linked by 44,338 citations.

- **Ego-Facebook** Leskovec & Mcauley (2012): A social network dataset representing ego-networks of 10 Facebook users, encompassing 4,039 nodes (friends) connected by 88,234 links. The dataset includes 193 ground-truth communities ("circles") manually labeled by the ego users, with an average of 19 circles per ego-network, each containing approximately 22 friends.

- **Wiki**[1]: A network of Wikipedia pages, consisting of 2,405 pages interconnected by 17,981 hyperlinks, with pages categorized into 19 distinct classes.

- **CORA-Full** Bojchevski & Günnemann (2017): An extended version of the Cora dataset, featuring 19,793 scientific publications classified into 70 categories. Each publication is represented by a binary word vector indicating the presence or absence of 1,433 unique words from the abstracts, with 65,311 citation links connecting the publications.

---

[1]`https://github.com/thunlp/MMDW/` (accessed July 28, 2023)

All aforementioned datasets are utilized in our link prediction experiments. For node classification tasks, we exclude the Ego-Facebook dataset due to the absence of node labels.

To benchmark our method's performance, we conduct comparative analyses against state-of-the-art graph embedding techniques, including LINE, SDNE, struc2vec, DeepWalk, and Node2vec. Our evaluation metrics focus on accuracy and macro F1-scores for both link prediction and node classification tasks, providing a comprehensive assessment of our method's capabilities in capturing both local and global graph structures.

## 5.2 LINK PREDICTION

Link prediction is a fundamental task in graph analysis that assesses the model's ability to capture the structural properties of the graph. In this study, we evaluate the performance of our reduced complexity Force-Directed Graph Embedding method on link prediction tasks across various datasets, focusing on the effects of key parameters: $m$, $k$, and $d$.

For the link prediction task, we employed a rigorous experimental protocol to ensure robust and unbiased evaluation of our method. The dataset was partitioned into training and test sets with a balanced 50:50 ratio, ensuring a comprehensive assessment of the model's generalization capabilities. Both sets were carefully constructed to maintain an equal distribution of positive (existing) and negative (non-existing) edge samples, mitigating potential biases in the evaluation process.

To represent each edge in the embedding space, we utilized the Hadamard product of the embeddings of its corresponding nodes. This approach, widely adopted in graph representation learning literature Grover & Leskovec (2016); Perozzi et al. (2014), effectively captures the pairwise interactions between node features in the learned embedding space. Formally, for an edge $(u, v)$, its representation $\mathbf{e}_{uv}$ is computed as:

$$\mathbf{e}_{uv} = \mathbf{z}_u \odot \mathbf{z}_v \tag{15}$$

where $\mathbf{z}_u$ and $\mathbf{z}_v$ are the embeddings of nodes $u$ and $v$ respectively, and $\odot$ denotes the Hadamard (element-wise) product.

For the classification task, we employed a Random Forest classifier, known for its robustness and ability to capture complex, non-linear decision boundaries **?**. The classifier was trained on the edge representations derived from the training set and evaluated on the held-out test set. We used the implementation provided by the scikit-learn library Pedregosa et al. (2011), with hyperparameters optimized through cross-validation to ensure optimal performance.

This experimental setup allows for a fair comparison with baseline methods and provides a comprehensive evaluation of our Force-Directed Graph Embedding method's capability to capture structural information relevant to the link prediction task.

The parameter $m$, defined as $m = t \log n, t \in \{10, 20, ..., 100\}$, determines the number of randomly sampled nodes beyond the $k$-ball for force calculations. The $k \in \{1, 2, 3\}$ parameter defines the radius of the $k$-ball centered at each node $u$, effectively controlling the extent of local neighborhood considered in the embedding process. We used 3 levels of values Lastly, $d$ represents the dimensionality of the embedding space.

Figure 2 illustrates the impact of different values of $t$ on link prediction accuracy, precision, and recall, across different datasets for $k = 1, 2$, and 3. Each column of plots belongs to a specific value of $k$, with the x-axis representing $t$ and the y-axis showing the metric value. Different lines within each plot represent distinct datasets.

As observed in Figure 2 and incontrast to intuition, link prediction metrics generally improve with decreasing $m$ across all datasets, except Ego-Facebook. Ego-Facebook is the only graph among these that has one connected component, i.e. all its nodes are connected mutually.

Figure 3 shows a comparison of quality of embeddings generated by different methods in terms of link prediction accuracy. This figure shows that the Force-Directed graph embedding with the proposed complexity reduction technique can still maintain a competitive quality, with slight improvement over famous methods such as Node2vec.

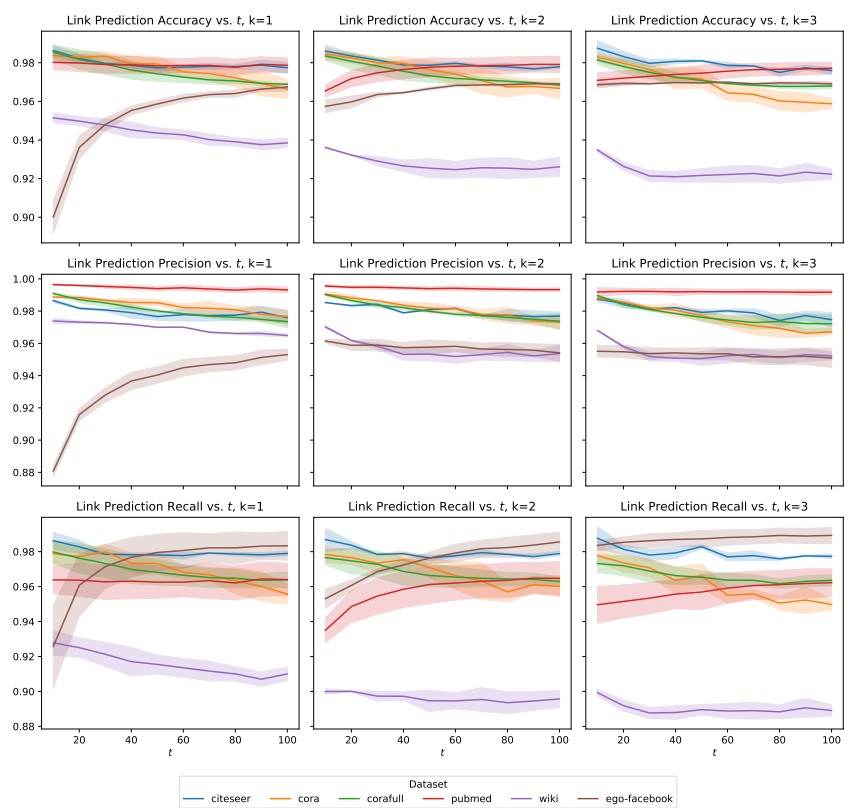

Figure 2: Effect of varying values of $t$ and $k$ on accuracy, precision, and recall of link prediction task on different datasets.

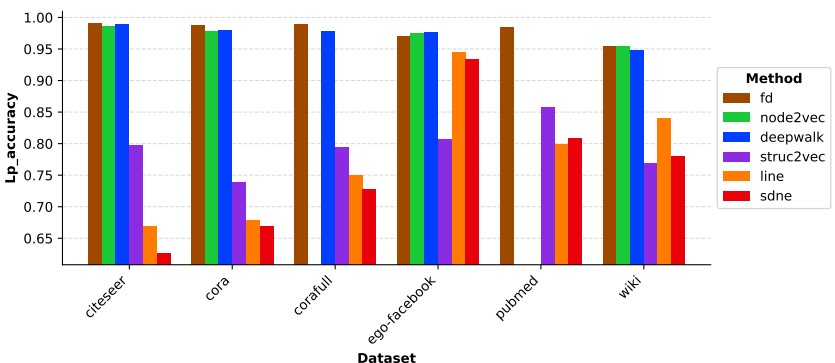

Figure 3: Comparison of link prediction accuracy against other methods.

## 5.3 NODE CLASSIFICATION

We used 50:50 train test to fit a random forest classifier. Figure 4 shows the node classification metrics over varying values of $t$ and $k$. Each column belongs to a specific value of $k$. According to this figure, the node classification metrics remained relatively consistent over different combinations of $t$ and $k$, with slight improvement over smaller values of $t$. Figure 5 shows a comparison of quality of embeddings generated by different methods in terms of node classification accuracy. This figure shows that the Force-Directed graph embedding with the proposed complexity reduction technique can still maintain a competitive quality, with slight improvement over famous methods such as Node2vec.

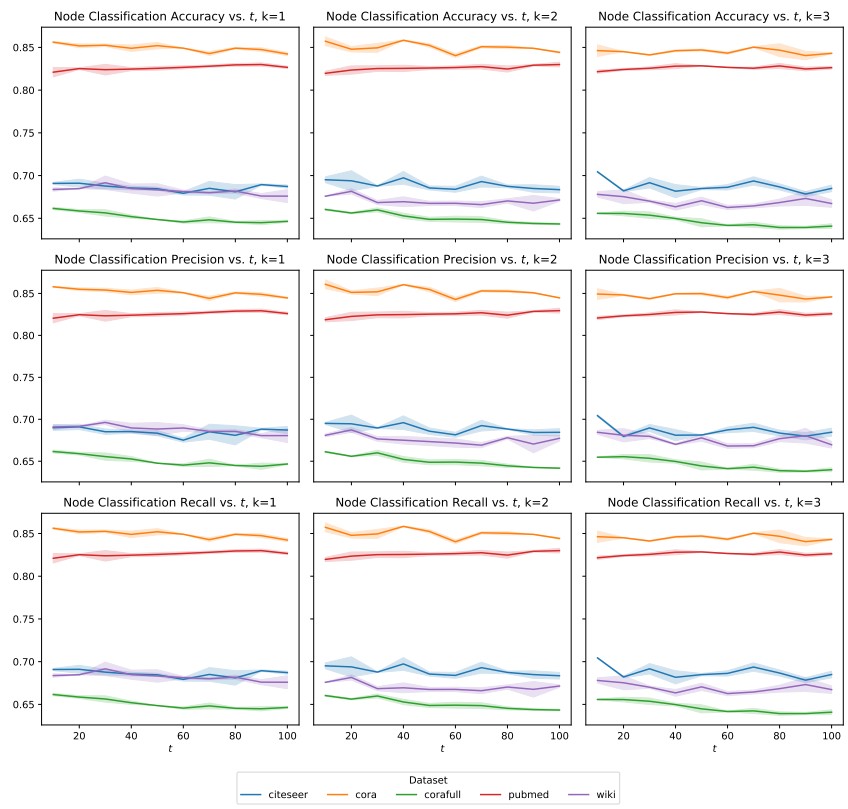

Figure 4: Effect of varying values of $t$ and $k$ on accuracy, precision, and recall of node classification task on different datasets.

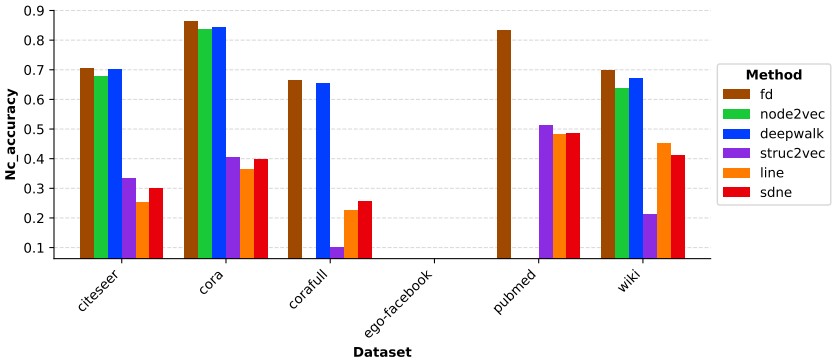

Figure 5: Comparison Figure

## 5.4 MEMORY UTILIZATION

To assess memory utilization, we calculate the percentage of non-zero elements in the hops matrix after keeping the entries that are used for calculating the corresponding forces. With an optimal implementation of the algorithm, it is possible to use a compact form of the matrices to calculate the forces. As depicted in 6, the percentage of memory utilization enhances with larger graphs (CORA-FULL, and PubMed), while maintaininig the quality of the generated embedding at a competitive level.

## 6 DISCUSSION

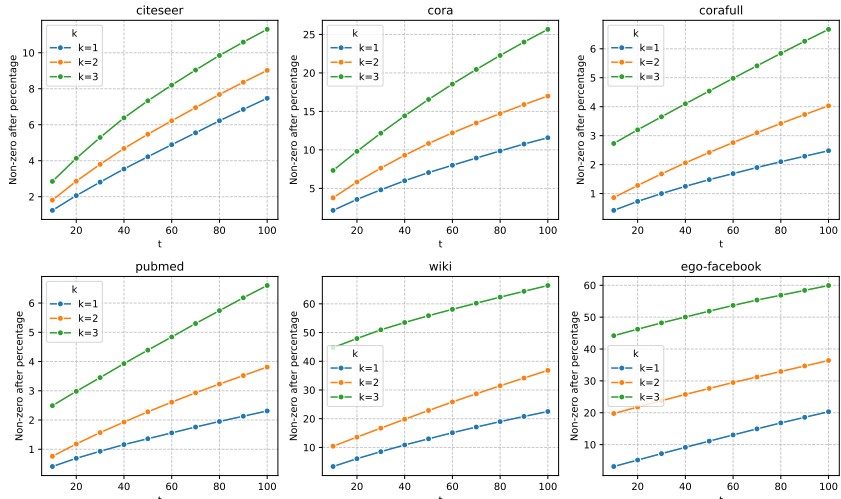

Figure 6: Memory utilization of the Force-Directed graph embedding algorithm with reduced complexity over different values of $t$ and $k$ .

!!! TO BE ELABORATED for CAMERA READY !!!

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
