# OpenReview forum: "Reducing Complexity of Force-Directed Graph Embedding"
_ICLR.cc/2025/Conference — Submitted to ICLR 2025_

### Official Review · Reviewer_exGs · 2024-10-17

**Soundness:** 3
**Presentation:** 2
**Contribution:** 2
**Rating:** 1
**Confidence:** 4

**Summary:**

The paper presents a force-directed graph embedding method that reduces the computational complexity of an earlier approach proposed by Lotfalizadeh et al. The authors introduce a modification to limit the force computations to $k$-hop neighborhoods and a few randomly sampled nodes, resulting in a reduction from $O(n^2)$ to $O(n \log(n))$ complexity. This makes the proposed method potentially more scalable for large graphs while maintaining competitive performance in unsupervised graph embedding tasks like link prediction and node classification.

**Strengths:**

(++) **Scalability Improvement**: The proposed complexity reduction from $O(n^2)$ to $O(n \log(n))$ is a notable improvement.

(++) **Practical Utility**: The paper demonstrates comparable or slightly better performance to some state-of-the-art graph embedding methods while offering scalability improvements, which suggests that the proposed approach has practical utility for large graph datasets.

**Weaknesses:**

(----) **Limited Novelty**: The main contribution is an incremental improvement to the original method by Lotfalizadeh et al. The use of $k$-hop neighborhoods and stochastic sampling for complexity reduction, while useful, does not represent a fundamentally new idea in the context of graph representation learning. The paper offers no new theoretical contributions, insights, or analyses.

(----) **Relationship to Previous Work**: The relationship to previous work, by Lotfalizadeh et al. (2023, 2024) is ambiguous. It is not clear how this work fundamentally extends the original force-directed embedding approach from these works.

(--) **Limited Evaluation and Analysis**: The paper only evaluates the quality of the proposed embeddings using two downstream tasks: link prediction and node classification.

(---) **Presentation Issues**: There are multiple signs that the paper is incomplete. Some examples:
* Placeholders such as "!!! A PICTURE TO BE INSERTED for CAMERA READY !!!" at Line 239 and "!!! TO BE ELABORATED ON for CAMERA READY !!!" at Line 452. The Discussion section is empty!
* Typographical errors such as "topolofy" on Line 234 and starting the sentences on Line 186 with lowercase letters.
* Broken reference on Line 301.
* $\log(n^2)$ in Line 027 in the abstract should be $O(n^2)$.
* The notation $\mathbf{z}_{uv} = \mathbf{z}_v - \mathbf{z}_u$ was introduced on Line 140 to facilitate brevity, then used in equation 3, not used in equations 8 and 9, then used again in equations 13 and 14.
* The paper mentions several well-known graph embedding techniques on Line 273, such as LINE, SDNE, DeepWalk, and Node2vec, but does not provide proper inline citations for them.

(--) **Marginal Performance Improvement**: While not a deal breaker, the downstream task performance improvement on previous methods is marginal at best, as can be seen in Figures 3 and 5.

**Questions:**

1. Could you clarify the differences between this paper and the previous work by Lotfalizadeh et al.?
2. Could you expand the empirical analysis and evaluation with more downstream tasks, e.g., multilabel classification or clustering?
3. Besides improved performance on downstream tasks, what desirable qualities do the FD embeddings have? E.g., the paper mentions reflecting the topology of the graph on Line 234 as a rationale for some of your choices. Would it be possible to evaluate that with metrics such as mean average precision?
4. Could you make your code available for reproducibility purposes?

---

### Official Review · Reviewer_nWvq · 2024-10-24

**Soundness:** 2
**Presentation:** 1
**Contribution:** 2
**Rating:** 3
**Confidence:** 5

**Summary:**

While this paper presents an interesting force-directed graph embedding approach, the manuscript feels incomplete. I recommend that the authors include more powerful baselines (e.g., DGI [1], GraphZoom [2]) and conduct evaluations on larger graphs (with over 1M nodes) to better demonstrate improvements in accuracy and scalability for their next submission.

[1] Veličković et al., "Deep graph infomax", ICLR'19 \
[2] Deng et al., "GraphZoom: A Multi-level Spectral Approach for Accurate and Scalable Graph Embedding", ICLR'20

**Strengths:**

NA

**Weaknesses:**

NA

**Questions:**

NA

---

### Official Review · Reviewer_3mBP · 2024-10-30

**Soundness:** 1
**Presentation:** 1
**Contribution:** 1
**Rating:** 3
**Confidence:** 4

**Summary:**

The paper presents a novel method for computing graph embeddings using a spring model without any neural network/model.

**Strengths:**

The overall idea is interesting and offers a complimentary perspective.

**Weaknesses:**

The paper seems incomplete.

**Questions:**

N/A

---

### Official Review · Reviewer_tLNK · 2024-11-01

**Soundness:** 2
**Presentation:** 1
**Contribution:** 2
**Rating:** 3
**Confidence:** 4

**Summary:**

The short review is as follows: the paper proposes a new set of graph embedding methods which instead of using message passing or back propagation, it uses spring model to construct graph embedding. Each nodes' positional embedding is the equilibrium state. I think there are quite a lot of paper that proposes new graph embedding methods, and in order to make a proposed method to work it needs to capture (1) global and local structure information (2) able to be learned and proactively adapted, otherwise no one would ever use the newly proposed embedding methods. From a brief walkthrough of the paper, I don't think the proposed method can be used as a way that proactively learns embeddings for nodes and graphs, which are useful for downstream tasks.

**Strengths:**

NA

**Weaknesses:**

NA

**Questions:**

NA

---

### Meta-Review · Area_Chair_R95j · 2024-12-10

**Metareview:**

This paper was submitted in a very incomplete state. I would like to discourage the authors from doing this in the future -- the review stage is not a place to get preliminary feedback on partial results, and by submitting incomplete papers like this, you do a disservice to the entire community, which is already plagued by high reviewer loads and difficulties in finding expert reviewers.

**Additional Comments On Reviewer Discussion:**

NA. Paper was incomplete and I just asked reviewers to give short reviews. No discussion with authors.

---

### Decision · Program_Chairs · 2025-01-22

Reject